# A deep learning AI model for determining the relationship between X-Ray detectors and patient positioning in chest radiography

**Bo Jiang, Junjiao Hu, Xiaofan Chen, Xiong Wu, Kai Deng, Haitao Yang, Weijun Situ ⓘ \*, Shan Jiang**

Department of Radiology, The Second Xiangya Hospital of Central South University, Changsha, Hunan, China

\* Stwj74@sina.com

## Abstract

### Purpose

The objective of this study was to create an artificial intelligence (AI) system capable of automatically detecting the positional relationship between an X-ray detector and the patient during anteroposterior chest radiography.

### Methods

In this study, a total of 22299 images depicting the positional states of X-ray detectors relative to patients were used to develop an AI system for the automatic determination of X-ray detector status. The images were captured from Routine clinical chest X-ray radiography practice settings, without exposing any patient privacy, adhering to the Declaration of Helsinki. The PyTorch library was utilized for customizing a Convolutional Neural Network (CNN) model for the training of the chest radiography positional determination model.

### Results

The average accuracy of Model A on the validation set was 0.9668, with an average loss function value of 0.1078. In contrast, Model B achieved an average accuracy of 0.9776, also with an average loss function value of 0.0970. In the test set results, Model A -fold 4 demonstrated a true negative rate (TNR) of 0.9925, negative predictive value (NPV) of 0.9925, precision of 0.9699, recall of 0.9700, accuracy of 0.9700, and an F1 score of 0.9699. Model B -fold 1exhibited a TNR of 0.9946, NPV of 0.9946, precision of 0.9787, recall of 0.9784, accuracy of 0.9784, and an F1 score of 0.9784. McNemar's test indicated statistical difference between the two models.

**Data availability statement:** All relevant data are within the manuscript and its Supporting Information files.

**Funding:** This work was supported by the following two grants: the Natural Science Foundation of Hunan Province (Joint Project of Science and Health), Award number: 2022JJ70139, and Clinical Medical Technology Innovation and Guidance Project In Hunan Province, Award Number: 2021SK53510. The funders had no role in study design, data collection and analysis, decision to publish, or preparation of the manuscript.

**Competing interests:** The authors have declared that no competing interests exist.

## Conclusion

The AI model utilizing a customized CNN architecture has demonstrated its potential to automatically detect the positional relationship between the patient and the X-ray detector during chest radiography procedures. This model can potentially alleviate the workload of radiologic technologists in producing chest radiographs and enhance the accuracy of the imaging process.

---

## Introduction

Chest radiography involves the process of X-rays passing through the chest and being projected onto film or flat-panel detectors to create an image. Chest X-rays can clearly record general pulmonary lesions while allowing for storage and comparative review. Moreover, the radiation dose received by patients is relatively low [1,2]. Given the rapid, simple, and cost-effective nature of X-ray imaging, chest radiography has become the preferred choice for routine thoracic examinations [3] and is a necessary component of health screenings. This has resulted in a high demand for chest X-ray imaging. Since the discovery of X-rays by Röntgen in 1895, X-ray imaging technology has progressed rapidly from traditional screen-film systems to computed radiography (CR) and further to digital radiography (DR) [2]. Additionally, modulation of tube current and voltage has advanced towards intelligent and low-radiation options [2,4] However, regardless of the evolution of X-ray machines, the process of chest radiography still requires radiologic technologists to operate the detectors with precision. This task is monotonous and labor-intensive, particularly in densely populated countries where the demand for chest X-ray examinations is significant, necessitating the use of numerous DR machines and technologists specifically for this purpose.

Therefore, reducing the workload of radiologic technologists and increasing the efficiency and accuracy of chest radiography remain critical challenges for the development of X-ray systems. With advancements in artificial intelligence (AI), computational capabilities have improved, leading to the rapid growth of deep learning and convolutional neural networks (CNN) in the field of computer vision [5,6]. AI-assisted technologies have begun to be applied to various medical challenges [7–10], and major DR manufacturers have incorporated numerous AI-driven intelligent positioning functions [8,11–15] However, these advancements have not effectively reduced the workload of technologists in chest X-ray imaging. Currently, DR's intelligent positioning technology does not decrease the working hours of technologists; it merely enhances the precision of their operations. Moreover, these intelligent systems are not dedicated to chest radiography, and there is no fully automated machine specifically designed for this purpose.

In response, we are developing a fully automated machine capable of accurately capturing chest X-rays, significantly reducing the workload of radiologic technologists. Technologists would only need to review and confirm the pre-imaging status of the chest X-rays or simultaneously review multiple DR images

via a network. This machine will automatically prompt patients to scan a code to input their examination information, demonstrate preparatory instructions via video, and after gaining patient confirmation, it will automatically open radiation shielding doors. It will guide patients into the designated imaging position and use visual recognition and intelligent voice prompts to further correct their posture. Once the positioning is confirmed, the machine will continuously adjust the position of the X-ray tube and detector, using additional visual recognition to finalize the optimal positioning. This state will then be swiftly reviewed by the technologist, who triggers the exposure. Upon completing the chest X-ray, the machine will open the shielding doors and indicate for the patient to exit. With this equipment, technologists would only need to approve the final imaging state, allowing one technologist to oversee multiple DR systems. This dedicated device will save considerable labor and prove especially beneficial in scenarios such as infectious disease outbreaks or military front lines.

The development challenge lies in accurately determining the relationship between the patient and the X-ray detector. Before the examination, the patient will be informed of the required posture via video. Once in the device, the machine will guide the patient to assume the correct position. When the posture is correct, the machine will ensure the detector locates the best imaging position from top to bottom. Our research focuses on developing a highly accurate AI model to assess whether the detector is too high, too low, correctly positioned, whether there is someone in the detector's field, and if the patient's posture is correct. Establishing this AI model will facilitate the creation of a fully automated machine for precise chest radiography.

## Materials and methods

### Study design

In this study, we utilized PyTorch library version 1.13 to customize a CNN architecture, aiming to develop an automated positional judgment model for chest radiography. This model is trained on a large dataset of images captured during Posteroanterior chest X-ray. The research protocol was approved by the Institutional Review Board (IRB) of The Second Xiangya Hospital of Central South University on 14 October 2021 (Approval No. 2021−761).All aspects involving participants in this study comply with the ethical standards of the National Research Council, and the principles of the 1964 Helsinki declaration and its later amendments or similar ethical standards. No patient's private parts or facial information was exposed, eliminating the need for additional informed consent.

### Data source

The data source for this study was exclusively the camera footage, which was deemed sufficient for judgment purposes. This approach is expected to considerably facilitate subsequent improvements to the X-ray equipment. We used a fixed-resolution (640x480 pixels) camera S-YUE X5(syue technology,Shenzhen,China)to record positional scenes of chest X-ray photography from February 20, 2023, to July 24, 2023, in a DR(DRX-Evolution, Rochester, America) room at Xiangya Hospital Central South University. All positioning scenes were conducted according to the Chinese standard [16] for Posteroanterior (PA) chest X-rays, with no supine or lateral shooting processes, reflecting real clinical examination scenarios. The camera was mounted directly above the X-ray tube, enabling real-time tracking of the status of the X-ray detector. The camera's field of view was limited to encompass the chest radiography X-ray detector panel and the vicinity, as illustrated in Fig 1. The video recordings excluded any patient-related privacy-sensitive areas and facial information, irrespective of gender and age differences. The collected videos were processed with Free Video to JPG Converter v.5.0.101 (DVDVideoSoft, Moscow,Russia), segmenting them into images at a rate of one frame per second. Images deemed irrelevant to the training model were discarded. Each image maintained the same resolution as captured by the camera, ensuring that the developed model could accurately determine the real-time status of the X-ray detector via the camera. A total of 22299 usable images were obtained for this purpose.

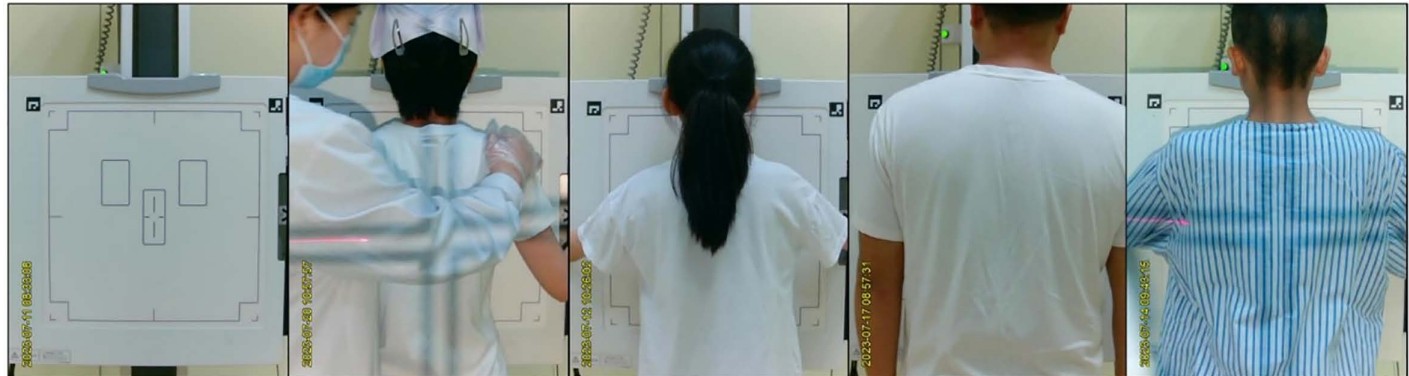

**Fig 1. Schematic diagram of the five categories of the positional relationship between the subject and the X-ray detector.** As shown in the figure, the video field of view only includes the chest radiography X-ray detector panel and its immediate vicinity, and is categorized into five classes based on its status(from left to right, the detector states are: "No one", "Others","Too High","Too Low" and "Suitable").

## Label classification

Two radiologic technologists, each with a minimum of eight years of professional experience, conducted the categorization of all the images suitable for training using a file-sorting methodology, and achieved consensus on the classifications [13]. The classification criteria strictly adhered to an established and rapid Posteroanterior chest X-ray method, which specifies that the detector's upper edge should extend approximately 3 cm above the shoulder peaks for optimal positioning [16]. The images were placed into five distinct folders: "Too High","Too Low","Suitable", "No one" and "Others"; The specific standards are as follows: "Too High" (detector upper edge is more than 2 cm above the standard position), "Too Low" (detector upper edge is more than 2 cm below the standard position), "Suitable" (detector upper edge is within 2 cm above or below the standard position, which is approximately 3 cm above the shoulder peaks), "No one" (no patient in the shooting area), and "Others" (The "Others" category mainly refers to situations where the patient appears in the detector area but has not reached the predefined central area or assumed the standard posture. This classification serves as a basis for developing a fully automated process for PA chest X-rays. In the subsequent development of the automatic shooting device, if the system determines an "Others" status, it will not adjust the detector's height but will first prompt the patient to assume the standard position before reassessing). with 4,174 in "Too High",1869 in "Too Low",3,915 in "Suitable", 6,331 in "No one",and 6,010 in "Others".The title of each folder corresponded to the classification label.

## Preprocessing steps

To develop an automated assessment model for the positioning of chest X-ray photography, 22299 images of chest radiograph positioning scenarios were utilized. In the preprocessing phase, algorithms were written using the OS libraries and glob libraries to read image classification labels. The skimage library was employed to adjust all image sizes to 100×100 pixels. The numpy library was then used to generate a dataset in.npy format, 3/5 of the data were used as the training set, 1/5 of the data were used for validation, employing K-fold cross-validation for data allocation and model training [17], and the remaining 1/5 of the data served as the test set. The specifics of the image preprocessing and data allocation are illustrated in Fig 2.

## Deep learning CNN architecture

Based on the introduction, it is evident that the automated process of posteroanterior chest X-ray photography requires visual recognition of five critical states. Therefore, we aimed to develop a precise five-classification AI model. Upon

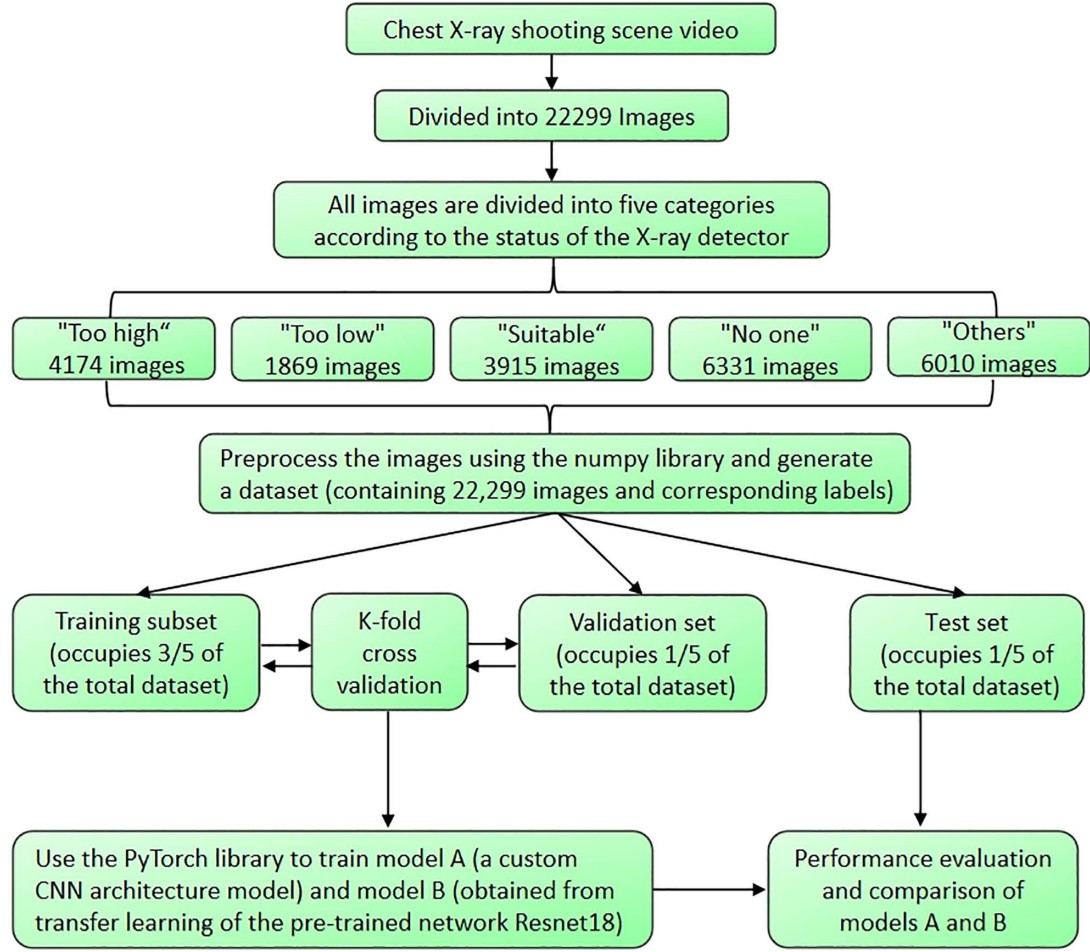

**Fig 2. Image Preprocessing and Data Allocation Flowchart.**

reviewing numerous journal articles [17–21], it was found that superior CNN models have been applied for assessing human standing postures [17]. Consequently, we created a custom CNN model, referred to as Model A, using the PyTorch library, drawing on previous experiences [17]. This CNN model consists of three convolutional layers, two pooling layers, and three fully connected layers. The first convolutional layer has a kernel size of 5×5, while the last convolutional layer has a kernel size of 3×3. The fully connected layers contain 1024, 512, and 5 nodes, respectively. On the other hand, traditional CNN models suffer from overfitting and vanishing gradient issues as the number of convolutional layers increases. Residual Network (ResNet) [20] is specifically designed to address these problems [21]. ResNet alleviates the vanishing gradient issue through residual connections and is widely used for feature extraction in medical imaging, reducing model complexity while enhancing computational efficiency, and has achieved significant progress in various aspects of medical image classification [20,21]. We chose ResNet18 over ResNet34/50 for two reasons: (1) Its parameter count (11.7M vs 21.8M/25.6M) is more suitable for real-time systems; (2) According to previous experiences, increasing network complexity in ResNet34/50 tends to lead to model overfitting [21]. Therefore, we employed ResNet18 to train Model B, aiming to explore more optimal visual recognition models. The detailed design of the convolutional neural network architecture and the model training process is illustrated in Fig 3. This marks the first exploration of using ResNet18 for training a five-classification model in the field of human standing posture recognition.

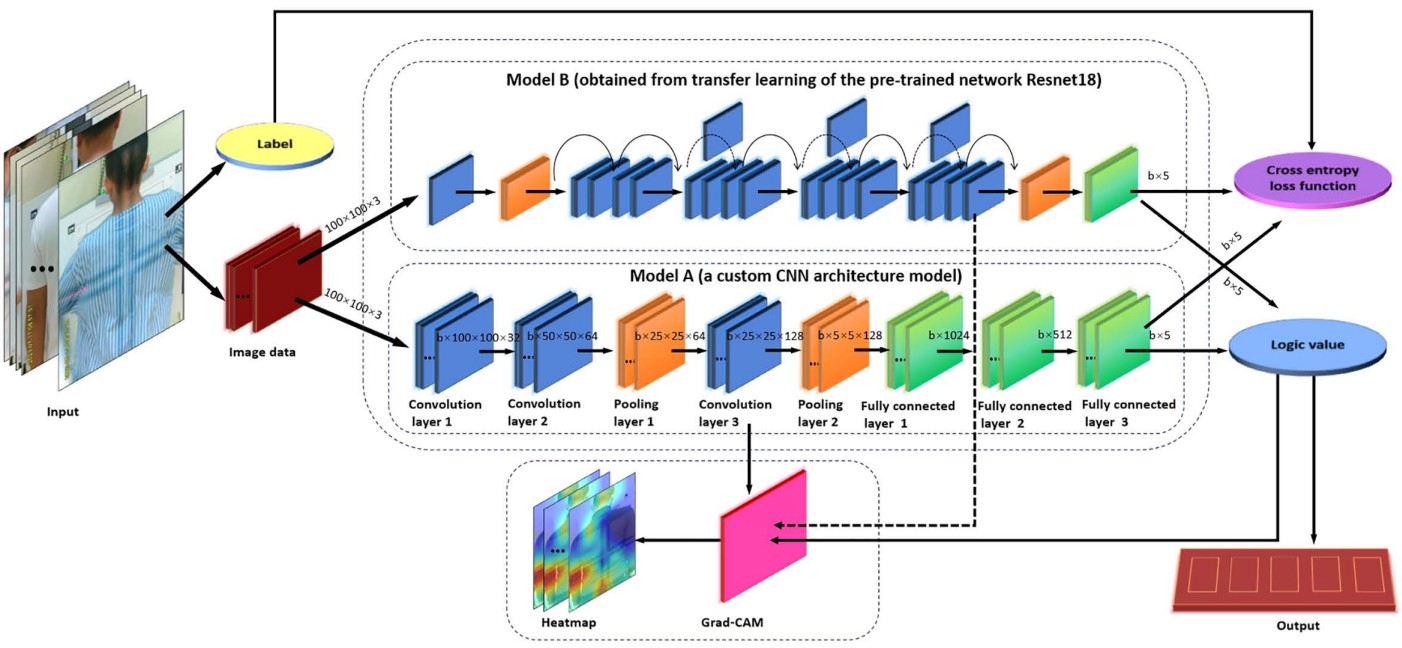

**Fig 3. Training Process for Two Types of Convolutional Neural Networks and AI Models.**

## Model training

An AI algorithm was developed utilizing the open-source programming language Python (version 3.9.21, Python Software Foundation, Wilmington, DE, USA). The AI model was generated using a custom CNN network through the PyTorch library, and supervised machine learning was applied for training the CNN model [22]. This training process utilized the computer equipment of the Post-processing Center of the Radiology Department at the XXX unit. Software settings and code development took place on a graphical computing server equipped with a passive GPU NVIDIA Tesla V100 16G (NVIDIA, California,USA). The training of the custom AI model was based on dynamic testing using the Adam adaptive learning rate optimizer [23], with an initial learning rate set at 0.0001. An L2 regularization function was used to prevent model overfitting, with a parameter value of 0.01, ensuring that the regularization did not induce sparsity while protecting against the excessive parameter growth [24]; Batch size for image input was set to 5 (lowering the batch size during testing led to increased accuracy of the model and a further reduction in the loss function, possibly due to the regularization preventing overfitting) [25]; Experimentation with iteration counts (epochs) ranging from 1 to 30 incorporated Early Stopping [26], to prevent model overfitting, ceasing when no improvement in validation set accuracy was observed and the loss function ceased to decrease. The final model was selected based on the lowest loss function value on the internal validation dataset and an accuracy rate close to that of the training set.

## Gradient-weighted class activation mapping (Grad-CAM)

Grad-CAM is a technique that allows for the visualization and localization of regions most instrumental in contributing to the predictive outcome, thereby making the neural network's decision-making process more interpretable [27]. In the present study, we defined a function to compute Class Activation Maps (CAM) based on the principles of Grad-CAM, which involves the utilization of global average pooling to compute the weights for each channel from the gradients of the final convolutional layer. These weights, when multiplied by the feature maps of the original image, yield a Class Activation Map in which each pixel value signifies the importance of that specific pixel region for the classification decision. By

applying the final model to compute the CAMs for images in the test set and subsequently merging them with the original images, we aimed to investigate the interpretability of the automatic positional judgement model for chest X-ray radiography. This process is instrumental in guiding the design of the size and quantity of convolutional layers for custom CNN models [27], we refrain from adding more convolutional layers if the weight outputs become too abstract or convoluted. The class-specific weight outputs of the ultimate convolutional layer are demonstrated in Fig 4.

## Statistical analysis

In order to objectively evaluate the advantages and disadvantages of the two models, k-fold cross-validation (CV) was employed for data allocation during training. Specifically, when k = 4, 3/4 of the data were used as the training set, while 1/4 served as the validation set. This process was repeated 4 times until each subset was used as the validation set. During model training, the accuracy and loss function of the validation set were output in real-time, and for each fold, the model with the highest accuracy was saved. The test set was then input into these models to obtain a batch of test data. The Confusion_Matrix subroutine from the scikit-learn 1.6.0 library was used to compute confusion matrices and derive related metrics. Based on these metrics, the optimal model from the four folds was selected as the final model. Once the best models for groups A and B were identified, statistical differences for each category were analyzed using the McNemar's subroutine from the statsmodels 0.14.4 library, with $P < 0.05$ indicating statistical significance and $P < 0.01$ indicating highly significant differences. Additionally, the confusion matrix and its derived metrics were calculated for each category in the best model, including sensitivity (true positive rate, TPR, equivalent to Recall), specificity (true negative rate, TNR), positive predictive value (PPV, equivalent to Precision), negative predictive value (NPV), accuracy (ACC), and F1 Score. ROC curves for the best models in groups A and B were generated using the Roc_Curve subroutine from scikit-learn 1.6.0, and the Area Under the Curve (AUC) was calculated. Calibration curves were generated using the Calibration_Curve subroutine, and the Brier score for each category was calculated using the Brier_Score_Loss subroutine.

## Results

Both Model A and Model B demonstrated significant effectiveness in accurately determining the spatial relationship between the chest X-ray detector and the patient. The accuracy and loss function values corresponding to the validation set for each fold of the models are presented in Table 1. The confusion matrices generated from the test set for each fold of the models, along with the derived metrics, can be found in Table 2. It can be observed from Table 2 that Model A achieved the highest accuracy and F1-score in fold-4, while Model B exhibited the best performance in accuracy and F1-score in fold-1. Further analysis of the confusion matrices for each subclass of the optimal models (Model A-fold-4 and Model B-fold-1) yielded the confusion matrix visualizations shown in Fig 5, The derived metrics obtained from the confusion matrices and the results of the McNemar's test are shown in Table 3. The ROC curves for each subclass of the optimal models are illustrated in Fig 6, the Calibration Curve along with the Brier Score is presented in Fig 7.

## Discussion

### Establishing an appropriate CNN model

As noted in the introduction, we are developing a fully automated machine for precise chest X-ray imaging. To enhance the intelligence of the examination process, it is crucial for the machine to accurately determine the positional relationship between the individual being examined and the X-ray detector. In the realm of automation technology, edge algorithms could be employed to discern this relationship [28]; however, they fail when the colors of the individual's clothing and the detector are similar. While techniques such as LIDAR or infrared detection could complement this deficiency [29,30], their integration with edge algorithms complicates the system, reduces stability, and significantly increases costs. Additionally, these methods do not assess whether the patient is in other specific states.

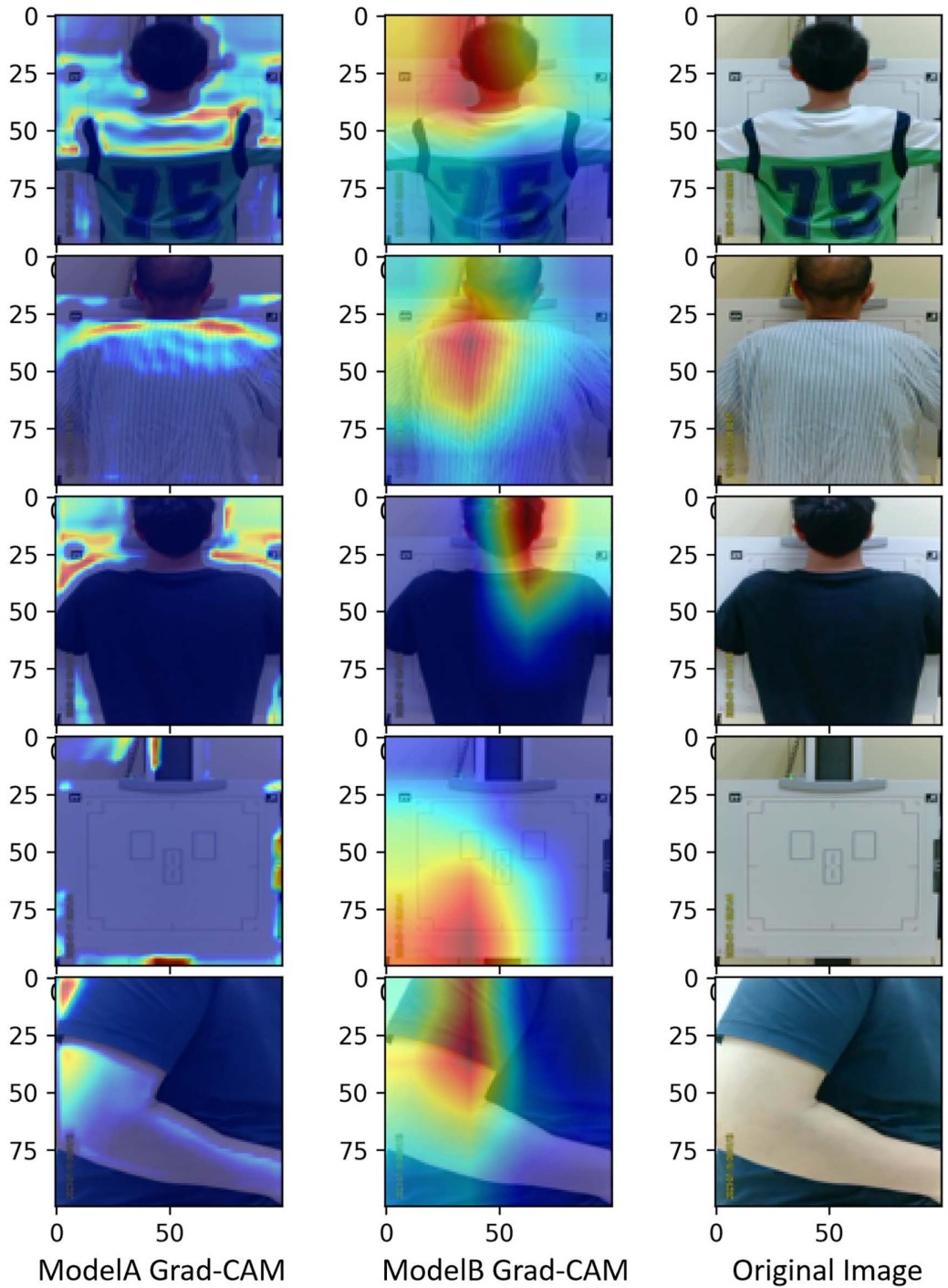

**Fig 4. Grad-CAM heatmaps generated for each test example in Model A (fold 4) and Model B (fold 1) compared with the original images (from top to bottom, each row corresponds to the detector states: "Too High","Too Low", "Suitable","No one" and "Others").**

**Table 1. Accuracy and loss function values for the validation set corresponding to each model fold.**

|  | Model A | | | | | Model B | | | | |
|---|---|---|---|---|---|---|---|---|---|---|
|  | fold1 | fold2 | fold3 | fold4 | Mean | fold1 | fold2 | fold3 | fold4 | Mean |
| Accuracy | 0.967 | 0.968 | 0.965 | 0.968 | 0.967 | 0.976 | 0.979 | 0.979 | 0.976 | 0.978 |
| Loss | 0.118 | 0.110 | 0.110 | 0.104 | 0.108 | 0.091 | 0.101 | 0.098 | 0.098 | 0.097 |

**Table 2. Derivative metrics from confusion matrices generated from the test set for each model fold.**

|  |  | TNR | NPV | Precision | Recall | Accuracy | F1 Score |
|---|---|---|---|---|---|---|---|
| Model A | fold1 | 0.9902 | 0.9902 | 0.9627 | 0.9606 | 0.9606 | 0.9610 |
|  | fold2 | 0.9913 | 0.9913 | 0.9655 | 0.9653 | 0.9653 | 0.9652 |
|  | fold3 | 0.9923 | 0.9923 | 0.9693 | 0.9691 | 0.9691 | 0.9690 |
|  | fold4 | 0.9925 | 0.9925 | 0.9699 | 0.9700 | 0.9700 | 0.9699 |
| Model B | fold1 | 0.9946 | 0.9946 | 0.9787 | 0.9784 | 0.9784 | 0.9784 |
|  | fold2 | 0.9935 | 0.9935 | 0.9740 | 0.9741 | 0.9741 | 0.9738 |
|  | fold3 | 0.9934 | 0.9934 | 0.9745 | 0.9738 | 0.9738 | 0.9739 |
|  | fold4 | 0.9927 | 0.9927 | 0.9708 | 0.9706 | 0.9706 | 0.9705 |

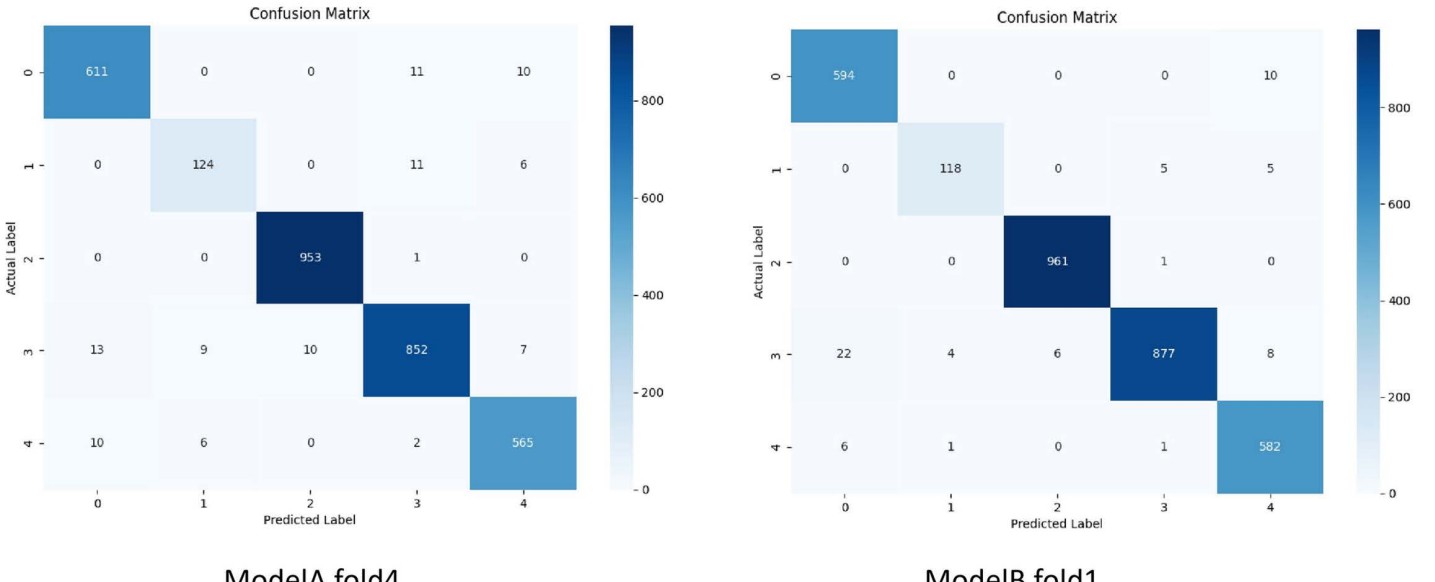

ModelA fold4                    ModelB fold1

**Fig 5. Confusion matrix visualizations generated for each category test in Model A (fold 4) and Model B (fold 1).**

Therefore, developing an accurate visual recognition model appears to be the optimal choice. Guided by the automation sequences, we identified five critical states as recognition criteria. The question then becomes: how to construct a suitable AI model for this five-class classification task? A review of the literature reveals the existence of superior CNN models for assessing human posture, which prompted us to adopt this approach to achieve comparable functionality. Model A, a custom CNN model we developed, is based on these findings. Notably, while traditional CNN models offer high accuracy, they suffer from overfitting and vanishing gradients as their convolutional layers increase. ResNet is designed to address these issues, and its use in pre-trained models has been rising in recent years. However, it has not been applied

**Table 3. Derivative metrics from confusion matrices and McNemar's test results for each category test in Model A (fold 4)and Model B (fold 1).**

|  |  | TNR | NPV | Precision | Recall | ACC | F1-Score | McNemar's test | P |
|---|---|---|---|---|---|---|---|---|---|
| Category1 | Model A | 0.9910 | 0.9918 | 0.9637 | 0.9668 | 0.9668 | 0.9652 | 5 | 0.7744 |
|  | Model B | 0.9892 | 0.9961 | 0.9550 | 0.9834 | 0.9834 | 0.9690 |  |  |
| Category2 | Model A | 0.9951 | 0.9944 | 0.8921 | 0.8794 | 0.8794 | 0.8857 | 3 | 1 |
|  | Model B | 0.9984 | 0.9968 | 0.9593 | 0.9219 | 0.9219 | 0.9402 |  |  |
| Category3 | Model A | 0.9955 | 0.9996 | 0.9896 | 0.9990 | 0.9990 | 0.9943 | 1 | 1 |
|  | Model B | 0.9973 | 0.9996 | 0.9938 | 0.9990 | 0.9990 | 0.9964 |  |  |
| Category4 | Model A | 0.9892 | 0.9832 | 0.9715 | 0.9562 | 0.9562 | 0.9638 | 14 | 0.1081 |
|  | Model B | 0.9969 | 0.9827 | 0.9921 | 0.9564 | 0.9564 | 0.9739 |  |  |
| Category5 | Model A | 0.9912 | 0.9931 | 0.9609 | 0.9691 | 0.9691 | 0.9650 | 4 | 0.1185 |
|  | Model B | 0.9912 | 0.9969 | 0.9620 | 0.9864 | 0.9864 | 0.9741 |  |  |
| Overall | Model A | 0.9925 | 0.9925 | 0.9699 | 0.9700 | 0.9700 | 0.9699 | 27 | 0.0203 |
|  | Model B | 0.9946 | 0.9946 | 0.9787 | 0.9784 | 0.9784 | 0.9784 |  |  |

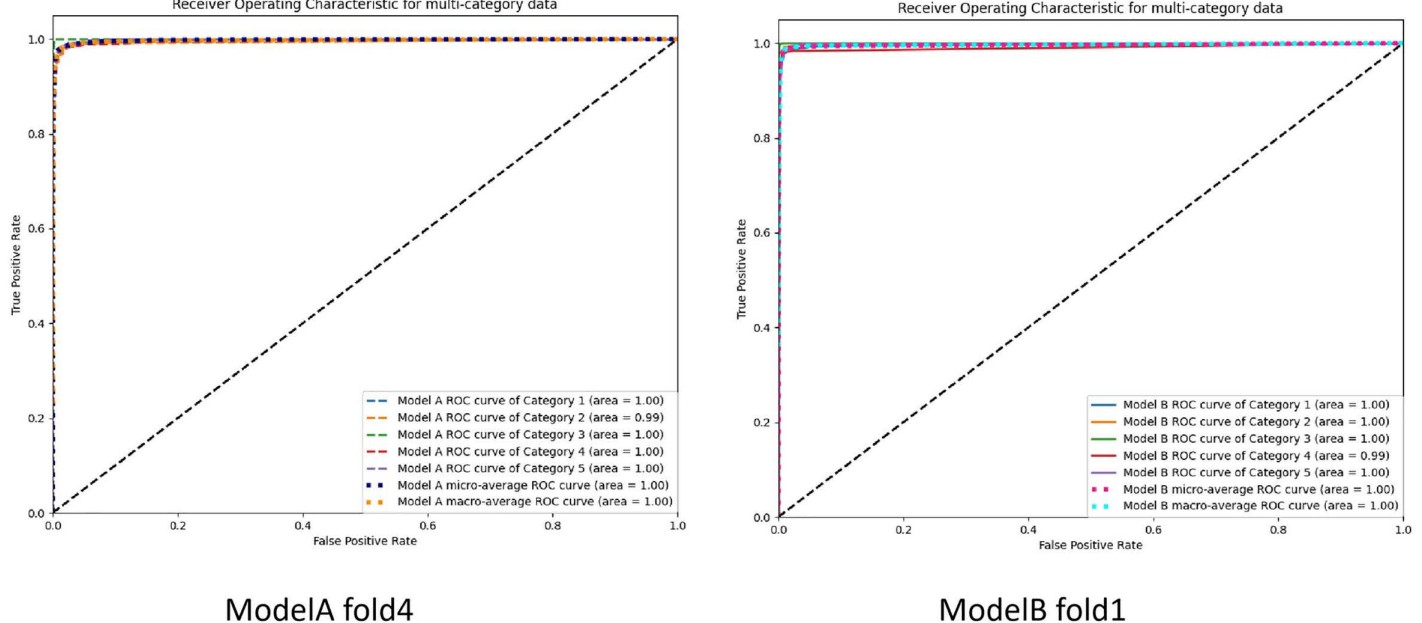

ModelA fold4        ModelB fold1

**Fig 6. ROC curves generated for each category test in Model A (fold 4)and Model B (fold 1).**

to determining the relative positions of humans and detectors. Therefore, we attempted to train Model B using ResNet18, and the results have been promising.

## Model design and tuning

Before model training, Two radiologic technologists conducted the categorization of all suitable images for training using a file-sorting methodology, The classification criteria strictly adhered to an established and rapid Posteroanterior chest X-ray method, which specifies that the detector's upper edge should extend approximately 3 cm above the shoulder peaks for optimal positioning [17],aiming to simulate real-world chest imaging scenarios. After several verifications by

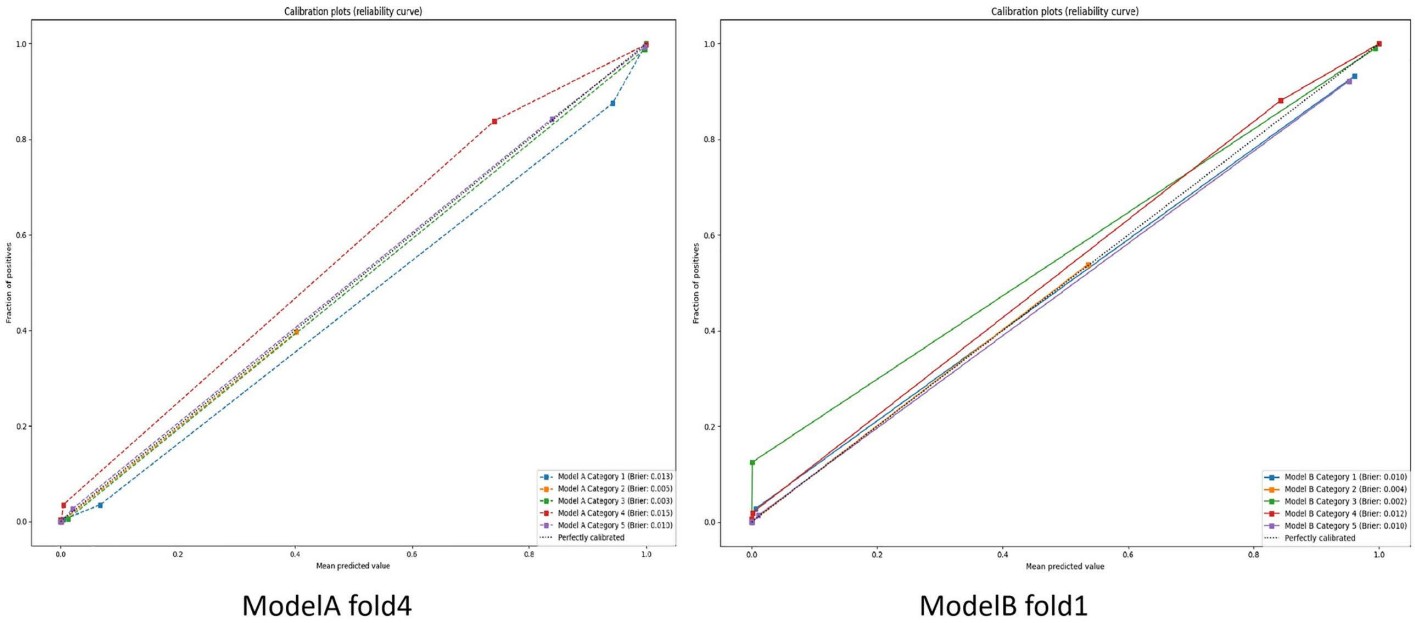

**Fig 7. Calibration curves and Brier scores generated for each category test in Model A (fold 4)and Model B (fold 1).**

two technologists, the images were divided into five categories as shown in Fig 1. This is because we found that patients could independently align themselves with the center of the detector following verbal cues, and feedback from these five states allows the detector to be correctly positioned. In terms of positioning accuracy, the Grad-CAM heatmaps for Model A (Fig 4) reveal that the model focuses on the intersection areas between the detector edges and the contours of the patient's shoulders. This focus aligns closely with the clinical standard used by radiologic technologists to determine whether the detector is positioned more than 3 cm above the shoulder peaks [16], indicating the model's decision-making is grounded in anatomically interpretable reasoning. Although the attention distribution for Model B appears more abstract, the residual structure effectively optimizes gradient flow, thereby enhancing classification performance (Table 3). Moreover, the machine consistently detected suitable positions for the detector from top to bottom during operation, stopping movement when the position was appropriate. Consequently, the category "Too Low" had fewer instances (1869) compared to other classifications, which aligns with the practical data distribution. Furthermore, we observed that the AI, after rigorous category learning, can still capture effective chest radiographs even with minor errors, as the vertical field range of most chest detectors is much larger than that of the lung field.

In the design of the convolutional neural network, excessive layers might lead to overfitting due to model complexity [27], Therefore, we utilized the Pytorch library to customize a previously optimized CNN training model (Model A) and employed a pre-trained residual network (ResNet18) for transfer learning to obtain Model B, aiming to prevent the decline in model accuracy caused by gradient vanishing; In addition to that, we employed optimized regularization functions and an appropriate number of iterations to avoid model overfitting during the training process of our custom AI model, L2 regularization helped to prevent overly large parameters without causing model sparsity [24], allowing us to set the batch size to 1, which further reduced the loss function while achieving higher accuracy. Early stopping was implemented to effectively prevent overfitting due to excessive training iterations [26]. To ensure model stability, training was based on the Adam optimizer with adaptive learning rates, with an initial rate of 0.0001 to avoid uncertainties from manual adjustments [23], In the exploratory process of hyperparameter tuning, other parameters attempted were found to be less effective than those of the final model.

## Performance analysis and interpretability analysis

Based on the data presented in Table 1, both Model A and Model B demonstrate high accuracy and low loss across each fold, but Model B has a lower average loss function. However, these metrics alone are insufficient to validate the models' generalization capabilities. To address this, we allocated one-fifth of the total data as a test set. As indicated in Table 2, the overall evaluation of the test set reveals that each fold of both Model A and Model B maintains high accuracy and F1 scores, with the ROC curves approaching perfection. This suggests a minimal degree of overfitting, attributable to rational data distribution, iterative hyperparameter tuning, and the implementation of L2 regularization and early stopping techniques.

Upon closer examination of Table 2, we observe that Model B exhibits slightly superior overall performance compared to Model A. This enhancement can be attributed to the application of the residual network, which shows statistical differences in the McNemar's test. The higher accuracy and F1 scores indicate that this model is likely to reduce errors in practical applications. Specifically, in the category testing detailed in Table 3, for Category 2, Model A (fold 4) achieved an accuracy of 0.8794 and an F1 score of 0.8857, while Model B (fold 1) reached an accuracy of 0.9219 and an F1 score of 0.9402. This discrepancy is also observable in the calibration curves, suggesting that the use of the residual network effectively mitigates the issue of the smaller data volume for Category 2, "Too Low" which is of considerable significance in real-world applications.

Interestingly, despite Model A's inferior performance in the classification predictions for Category 2, it excels in interpretability as demonstrated by the Grad-CAM heatmaps. As illustrated in Fig 4, the heatmap generated by Model A appears to align closely with human cognitive logic, accurately capturing areas of the detector beyond the human silhouette and increasing attention to the edges of both the human form and the detector. In contrast, the attention mechanism of Model B seems less comprehensible. Nonetheless, we believe that Model B possesses greater practical value overall.

## Limitations and future prospects

In summary, our project holds significant practical value. On one hand, inferring the spatial relationship between the X-ray detector and the patient solely based on camera status entails relatively low modification costs. On the other hand, applying this model to the development of fully automated precision chest X-ray machines can significantly reduce the workload and monotonous operating environment for radiologic technologists. However, this is not sufficient to eliminate the need for final review by technicians, as this study has the following limitations: 1) The data is sourced from a single center with fixed DR equipment (DRX-Evolution,Rochester,America), and cross-institutional generalizability needs to be verified; 2) The "Too Low" sample size is relatively small (n = 1,869), which may affect the performance of Category 2 classification (Table 3); 3) Pediatric/special body type patients were not tested. As part of an ongoing series of studies, we aim to address these issues through multi-center data transfer learning in the future. We will enhance the dataset for the "Too Low" category, expand the sample size, explore additional geometric markers or superior deep learning models to improve model specificity, and strive to achieve the goal of fully intelligent chest X-ray acquisition.

## Conclusion

This study presents the design and training of a usable deep learning convolutional neural network-based AI model to determine the positional relationship between the X-ray detector and the patient during chest radiography. The Model B's accuracy (0.9784) is remarkably promising, warranting only minimal corrective intervention from technicians for low-probability errors. Together with stepper motor linear modules for feedback-based adjustment, the developed system can appreciably reduce the workload associated with radiologic technologist positioning and tedious operational states, yet there remains room for optimization towards the full realization of automated chest radiograph acquisition.In the future, the implementation of fully automated DR chest radiography will significantly improve the allocation of resources for chest X-ray examinations in medical examination centers and will be applicable in special scenarios such as pandemics and battlefields.

## Highlights of the study

- To our knowledge, there has been no research on AI models that automatically detect the positional relationship between X-ray detectors and patients during chest radiography.

- This study designed and trained a usable artificial intelligence model through deep learning to determine the positional relationship between X-ray detectors and patients during chest radiography. The model achieved considerable accuracy (0.9784).

- We are in the process of developing a fully automated machine for precise chest X-ray acquisition, which has the potential to significantly alleviate the workload of radiologic technologists. Technologists will only need to verify the status prior to the X-ray acquisition or concurrently review images from multiple digital radiography (DR) machines via a network. This specialized equipment is expected to save substantial labor, proving particularly advantageous in unique work environments such as those encountered during infectious disease outbreaks and on the frontlines of conflict. The establishment of this AI model in this study will facilitate the advent of fully automated machines for precise chest X-ray imaging.

## Author contributions

**Conceptualization:** Junjiao Hu, Shan Jiang.

**Data curation:** Xiaofan Chen.

**Formal analysis:** Junjiao Hu, Xiong Wu.

**Investigation:** Xiong Wu.

**Methodology:** Kai Deng, Haitao Yang, Shan Jiang.

**Project administration:** Weijun Situ.

**Resources:** Weijun Situ.

**Validation:** Xiaofan Chen.

**Writing – original draft:** Bo Jiang.

**Writing – review & editing:** Bo Jiang, Haitao Yang.

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
