## [Decision Letter · Decision Letter 0]

11 Dec 2024

Dear Dr. 司徒,

Thank you for submitting your manuscript to PLOS ONE. After careful consideration, we feel that it has merit but does not fully meet PLOS ONE’s publication criteria as it currently stands. Therefore, we invite you to submit a revised version of the manuscript that addresses the points raised during the review process.

We look forward to receiving your revised manuscript.

Kind regards,

Ihssan S. Masad, PhD

Academic Editor

PLOS ONE

Journal Requirements:

6. Please ensure that you include a title page within your main document. You should list all authors and all affiliations as per our author instructions and clearly indicate the corresponding author.

Reviewers' comments:

Reviewer's Responses to Questions

**Comments to the Author**

1. Is the manuscript technically sound, and do the data support the conclusions?

Reviewer #1: Partly

Reviewer #2: Partly

2. Has the statistical analysis been performed appropriately and rigorously?

Reviewer #1: Yes

Reviewer #2: No

3. Have the authors made all data underlying the findings in their manuscript fully available?

Reviewer #1: No

Reviewer #2: Yes

4. Is the manuscript presented in an intelligible fashion and written in standard English?

Reviewer #1: Yes

Reviewer #2: Yes

Reviewer #1: General:

The introduction needs to be reworked. There are some irrelevant details (see comments) and some missing information. For instance, why no background on CNN or more detailed discussion of prior image classification work? Also, the purpose/approach of the study should be clarified.

Was there any attempt to measure distance to participants or otherwise quantify their position in space? This seems important and is not addressed.

What are other alternatives to this approach? I think this paper is missing any comparison to current work. Pose estimation is quite good these days and there are likely off-the-shelf products that could aid in real-time positioning of a patient. One could even imagine a simple edge detection algorithm could be used here. Your approach is reasonable in a vacuum, but the paper is missing justification and context, especially considering that the only measure used is whether or not the patient is “too high” or “too low”.

The results section is limited. It seems to be missing a reference to the Grad-CAM results. There is not anything on the training or validation of the model. I would be interested in seeing some of the testing that went in to selecting your hyper-parameters too.

I would include baseline example images of each category. Figure one has images that are unlabeled, and it is not clear what is what.

The figures in general seem a bit blurry and do not have good/description legends.

I have somewhat of a problem with the simple categories. They are very limited. What is too high? What is too low? How is that actually determined (outside of expert opinion)? What about distance from patient? Posture? Etc. Is that the only issue that is present? What if someone is just a little too high or a little too low? These categories are presented without good context. I also wonder if your categories correlate at all with end image quality (the actual x-rays). This is very important I think for the practical application and it is not addressed here.

Specific:

Lines 36 – 40: Why do you mention fluoroscopy here? I don’t see how this section fits into the introduction. Fluoroscopy is a separate technique that would not necessarily be used in place of chest x-ray (or vice versa). I don’t see the reason for the comparison. This section does not add anything of value. I would suggest removing it or rewriting it to better integrate with the paper. I would rather see background about where and how x-ray might be used.

Lines 41-48: This feels like a better start to the introduction and is more relevant to the research question. I would consider integrating this paragraph and the first to improve flow.

Lines 45-48: I would like to see some citations here. Has there been prior work done? Why is this an issue? Please expand more here.

Lines 55-59: I think that you need to clarify that this modeling is attempting to use external camera data to aid in positioning. It is not obvious.

Lines 83 – 84: Was the classification solely based on visual distance/positioning? I would think that the resulting x-ray should be the standard here i.e., that the images are graded based on the clarity/positioning/quality of the end result image. Why was this not done?

Line 87: Please specify what the “No-one” and “Others” classifications mean.

Lines 94-95: Why such a comparatively small test set?

Line 99: How is this different from conventional models? Are you referring to a particular architecture? A CNN with 2 convolutional layers and 3 fully connected layers does not strike me as unusual.

Line 115: I am surprised by this low batch size produces the best results. It isn’t unreasonable, it is just lower than usual (for a “low” batch size).

Lines 132-134: What was the result here? Did you make any changes to the model? The referenced figure is interesting, but it isn’t obvious to me what you did with that info.

Lines 154-156: It seems that this is repeated from table 1. I do not think the information needs to be in both places. The table doesn’t really add much. Additionally, it is confusing to read, at a minimum I would add acronyms to the legend.

Line 158: Why is this section not in the intro? I don’t think it belongs in the discussion.

Line 169: This entire subsection is a mishmash of information that belongs elsewhere. The Grad-CAM results should be in the results section. The methods should mention that you tested multiple configurations of convolutional layers. It is not clear what is considered a well-positioned x-ray.

Line 201: “the TNR of 0.95.6” should this be “0.956”?

Lines 208-209: This seems to be a major limitation in the training set. I would be interested to know what the performance was for each category. For example, what was the accuracy of each category? Were there discrepancies?

Reviewer #2: Your manuscript is technically strong and provides valuable data to support its conclusions. However, there are a few areas where refinements could further enhance the clarity, rigor, and practical relevance of the work.

*Model validation

The reported accuracy metrics are impressive, but the validation and test sets are relatively small compared to the training data. It would be helpful to discuss whether the results are generalizable to broader populations or different imaging setups. Additionally, consider addressing how the model might perform in varied real-world scenarios, such as with different equipment or patient positions, to better illustrate its robustness.

*Statistical analysis

Your study employs solid and appropriate statistical methods, but there are a few areas where additional details could enhance the clarity and reliability of the results.

Confidence Intervals (CIs): Including confidence intervals for key performance metrics like accuracy, sensitivity, specificity, and precision would provide a clearer picture of how consistent and reliable these metrics are. Confidence intervals offer insight into the potential variability of your results and allow readers to assess how robust your findings are in different scenarios.

Class imbalance: The imbalance in image categories, such as "Too Low," could potentially affect the model’s performance metrics. To address this, consider reporting class-specific metrics, such as precision, recall, and F1-scores for each category. This will help highlight any discrepancies in performance across the different classes and ensure transparency about how well the model handles underrepresented categories.

Statistical comparisons: If you are comparing the performance of your model to other baseline methods or alternative approaches, it would be beneficial to include statistical significance testing. For instance, using McNemar’s test for paired categorical data can help demonstrate whether differences in performance are statistically meaningful. This will make your claims about the model’s superiority more compelling.

Model calibration: Providing an evaluation of your model’s calibration, such as a calibration curve or metrics like the Brier score, would be particularly useful for readers. This would show how well the predicted probabilities align with actual outcomes, which is critical in clinical applications where confidence in predictions can directly impact patient care.

Variability assessment: Consider employing techniques like bootstrapping or cross-validation to assess the variability of your metrics across different data subsets. This would provide a more comprehensive view of your model’s generalizability and reliability, helping readers understand how it might perform in broader applications.

*Presentation and Writing

Overall, the manuscript is well-written, but some sections, particularly the methodology, feel overly dense. Simplifying the language and structure in these areas would make the content more accessible to a wider audience. Additionally, figures and tables are useful, but captions for Figure 3 (training process) and Figure 4 (Grad-CAM) could be expanded to better explain their relevance and key takeaways.

*Discussion of limitations

The discussion does not fully explore some of the limitations of the study. For example, the imbalance in image categories, such as "Too Low," may affect the model’s performance. It would be beneficial to address this issue and propose strategies for handling similar imbalances in future research.

*Future directions and practical implications

Your manuscript hints at the practical implications of the AI model, such as reducing the workload of radiologic technologists. Expanding on this by discussing challenges related to clinical integration—like addressing ethical considerations and providing user training—would add depth to the discussion. This would help readers better understand how your model could be implemented in real-world clinical settings.

**Do you want your identity to be public for this peer review?** For information about this choice, including consent withdrawal, please see our Privacy Policy

Reviewer #1: No

Reviewer #2: No

---

## [Author Response · Author response to Decision Letter 1]

17 Feb 2025

Dear Editor and Reviewers of the esteemed journal,

Greetings!

We appreciate the acknowledgment of our work by the editorial team and reviewers, and we have found your comments highly beneficial. We sincerely believe that these suggestions will help us present our research results and academic viewpoints more effectively. Below are our detailed responses to the revisions made to the manuscript. Please allow us to highlight our responses in distinct colors for quick reference.

Journal Requirements:

Response: Revisions have been made according to the official templates provided, although minor discrepancies may exist due to the PDF format. We are happy to revise further if needed.

Response: Author information has been revised according to the format.

Response: We confirm that the funding grant numbers are accurate, and intergovernmental certification has been provided.

Response: We confirm that the results related to the content of the article are included in the manuscript. As for the specific dataset used to train the model, it is not a public dataset and is too large to provide at present. Interested readers may contact jiangbo@csu.edu.cn for inquiries.

Response: We have completed the association of the ORCID iD as required.

6. Please ensure that you include a title page within your main document. You should list all authors and all affiliations as per our author instructions and clearly indicate the corresponding author.

Response: A new title page has been uploaded for your review.

We thank the reviewers for their thorough suggestions, which prompted us to reflect deeply on our manuscript. Initially, we did not elaborate on why we established such a model; in fact, our research aims to develop a fully automatic chest X-ray machine that can significantly reduce the workload of radiological technicians in populous countries. Especially in medical examination centers where many examinees require long waiting times, and in infectious disease zones or warfronts, automated examinations can help avoid various issues. The five-category classification of the model is designed as a critical decision node for this automation. We have continuously optimized this aspect of our work, and based on the reviewers' comments, we have incorporated recent progress into the manuscript to better conform to academic paper standards and form robust comparisons. Accordingly, we have extensively revised and optimized the dataset, experimental design, and statistical analysis. The following are detailed responses to the individual issues raised.

Reviewer #1: General:

The introduction needs to be reworked. There are some irrelevant details (see comments) and some missing information. For instance, why no background on CNN or more detailed discussion of prior image classification work? Also, the purpose/approach of the study should be clarified.

Response: The introduction has been rewritten, detailing why the model was established, the rationale behind the chosen classifications, and how our work builds upon previous studies while explaining the reasons for adopting this method.

Was there any attempt to measure distance to participants or otherwise quantify their position in space? This seems important and is not addressed.

Response: Apologies for not detailing this in the initial submission. Our spatial distances strictly adhere to the imaging methods described in literature, both in terms of shooting distance and positioning techniques, which are standard practices in China for chest X-rays.

What are other alternatives to this approach? I think this paper is missing any comparison to current work. Pose estimation is quite good these days and there are likely off-the-shelf products that could aid in real-time positioning of a patient. One could even imagine a simple edge detection algorithm could be used here. Your approach is reasonable in a vacuum, but the paper is missing justification and context, especially considering that the only measure used is whether or not the patient is "too high" or "too low".

Response: Following your suggestion, we have added a detailed discussion in the first paragraph of the Discussion section regarding alternative approaches, why AI was chosen over other positioning methods, and the criteria for measurement.

The results section is limited. It seems to be missing a reference to the Grad-CAM results. There is nothing on the training or validation of the model. I would be interested in seeing some of the testing that went into selecting your hyperparameters too.

Response: As suggested, descriptions and discussions of Grad-CAM have been added.

I would include baseline example images of each category. Figure one has images that are unlabeled, and it is not clear what is what.

The figures in general seem a bit blurry and do not have good/descriptive legends.

I have somewhat of a problem with the simple categories. They are very limited. What is too high? What is too low? How is that actually determined (outside of expert opinion)? What about distance from patient? Posture? Etc. Is that the only issue that is present? What if someone is just a little too high or a little too low? These categories are presented without good context. I also wonder if your categories correlate at all with end image quality (the actual x-rays). This is very important I think for practical application and it is not addressed here.

Response: Your points are well taken. We have strict reference standards that Chinese radiological technicians follow. "Too high" and "too low" refer to positions relative to the correct method, which stipulates that the detector's upper edge must be 3cm away from the examinee's acromion on both sides. This ensures batch processing and standardization of chest X-rays.

Specific:

Lines 36 – 40: Why do you mention fluoroscopy here? I don’t see how this section fits into the introduction. Fluoroscopy is a separate technique that would not necessarily be used in place of chest x-ray (or vice versa). I don’t see the reason for the comparison. This section does not add anything of value. I would suggest removing it or rewriting it to better integrate with the paper. I would rather see background about where and how x-ray might be used.

Response: Following your suggestion, we have removed the content related to fluoroscopy. Initially, we compared chest examination methods before the advent of DR as a common practice in large populations. However, with rapid technological development, comparing fluoroscopy no longer holds much relevance.

Lines 41-48: This feels like a better start to the introduction and is more relevant to the research question. I would consider integrating this paragraph and the first to improve flow.

Response: We have thoroughly integrated the introduction and discussion sections to improve the flow.

Lines 45-48: I would like to see some citations here. Has there been prior work done? Why is this an issue? Please expand more here.

Response: We have added references to similar studies as per your suggestion. Our CNN architecture was designed based on one of the highly feasible networks from previous studies, although papers exactly matching our model's application are scarce.

Lines 55-59: I think that you need to clarify that this modeling is attempting to use external camera data to aid in positioning. It is not obvious.

Response: We have specifically described in the Methods section that the data originates from scene videos captured by a camera fixed above the X-ray tube.

Lines 83 – 84: Was the classification solely based on visual distance/positioning? I would think that the resulting x-ray should be the standard here i.e., that the images are graded based on the clarity/positioning/quality of the end result image. Why was this not done?

Response: Your point is well taken. We adhere to strict reference standards followed by Chinese radiological technicians. "Too high" and "too low" refer to positions relative to the correct method, which stipulates that the detector's upper edge must be 3cm away from the examinee's acromion on both sides, aiding in batch processing and standardization of chest X-rays.

Line 87: Please specify what the “No-one” and “Others” classifications mean.

Response: In the introduction, we have added descriptions. The model provides visual recognition for an automated shooting machine under development. If the examinee has not reached the detector area, video and voice prompts guide them. Therefore, detecting the absence of a person ("No-one") and identifying incorrect postures ("Others") are crucial for guiding proper positioning.

Lines 94-95: Why such a comparatively small test set?

Response: Taking your advice, we have reallocated the dataset according to similar studies, ensuring the test set comprises 1/5 of the total data.

Line 99: How is this different from conventional models? Are you referring to a particular architecture? A CNN with 2 convolutional layers and 3 fully connected layers does not strike me as unusual.

Response: Our customized CNN architecture is modeled after previous studies because this configuration has achieved high accuracy in judging standing postures.

Line 115: I am surprised by this low batch size producing the best results. It isn’t unreasonable, it is just lower than usual (for a “low” batch size).

Response: Using the smallest batch size aimed to maximize the effectiveness of the limited training set. We verified that batch sizes between 1 and 5 showed no significant changes in accuracy, so using a batch size of 5 can expedite training and hyperparameter tuning.

Lines 132-134: What was the result here? Did you make any changes to the model? The referenced figure is interesting, but it isn’t obvious to me what you did with that info.

Response: We have detailed statistical processing and included K-fold cross-validation, McNemar’s test for statistical differences, confusion matrices, calibration curves, and Brier scores.

Lines 154-156: It seems that this is repeated from table 1. I do not think the information needs to be in both places. The table doesn’t really add much. Additionally, it is confusing to read, at a minimum I would add acronyms to the legend.

Response: We have used K-fold validation to manage data and clarified three tables to present the data succinctly.

Line 158: Why is this section not in the intro? I don’t think it belongs in the discussion.

Response: We have comprehensively integrated the introduction and discussion sections.

Line 169: This entire subsection is a mishmash of information that belongs elsewhere. The Grad-CAM results should be in the results section. The methods should mention that you tested multiple configurations of convolutional layers. It is not clear what is considered a well-positioned x-ray.

Response: We have added discussions on Grad-CAM and introduced an experimental group with residual networks, finding that our customized CNN offers better interpretability.

Reviewer #2:

Your manuscript is technically strong and provides valuable data to support its conclusions. However, there are a few areas where refinements could further enhance the clarity, rigor, and practical relevance of the work.

Response: Thank you for acknowledging our work. We have added new experimental groups, restructured the dataset, and employed the recommended statistical methods to evaluate the model's performance.

*Model validation

The reported accuracy metrics are impressive, but the validation and test sets are relatively small compared to the training data. It would be helpful to discuss whether the results are generalizable to broader populations or different imaging setups. Additionally, consider addressing how the model might perform in varied real-world scenarios, such as with different equipment or patient positions, to better illustrate its robustness.

Response: Following your advice, we have reallocated the test set, ensuring 1/5 of the data is used for testing. We also increased testing across various categories and discussed the model's generalizability.

*Statistical analysis

Your study employs solid and appropriate statistical methods, but there are a few areas where additional details could enhance the clarity and reliability of the results.

Confidence Intervals (CIs): Including confidence intervals for key performance metrics like accuracy, sensitivity, specificity, and precision would provide a clearer picture of their consistency and reliability. Confidence intervals offer insight into potential variability and allow readers to assess the robustness of findings in different scenarios.

Class imbalance: The imbalance in image categories, such as "Too Low," could potentially affect the model’s performance metrics. To address this, consider reporting class-specific metrics, such as precision, recall, and F1-scores for each category. This will highlight any discrepancies and ensure transparency regarding underrepresented categories.

Statistical comparisons: If comparing the model's performance to other baseline methods or alternative approaches, including statistical significance testing would be beneficial. For example, using McNemar’s test for paired categorical data can demonstrate meaningful performance differences.

Model calibration: Providing an evaluation of model calibration, such as a calibration curve or Brier score, would show how well predicted probabilities align with actual outcomes, critical in clinical applications.

Variability assessment: Techniques like bootstrapping or cross-validation can assess metric variability across different data subsets, offering a comprehensive view of generalizability and reliability.

Response: We have added experimental groups and recalibrated all models. Statistical analyses now include McNemar’s tests, confusion matrices, calibration curves, and Brier scores. We have also clarified the criteria fo

---

## [Decision Letter · Decision Letter 1]

23 Jun 2025

Dear Dr. 司徒,

Thank you for submitting your manuscript to PLOS ONE. After careful consideration, we feel that it has merit but does not fully meet PLOS ONE’s publication criteria as it currently stands. Therefore, we invite you to submit a revised version of the manuscript that addresses the points raised during the review process.

We look forward to receiving your revised manuscript.

Kind regards,

Ihssan S. Masad, PhD

Academic Editor

PLOS ONE

Reviewers' comments:

Reviewer's Responses to Questions

**Comments to the Author**

Reviewer #1: All comments have been addressed

Reviewer #3: (No Response)

Reviewer #4: (No Response)

2. Is the manuscript technically sound, and do the data support the conclusions?

Reviewer #1: Yes

Reviewer #3: Partly

Reviewer #4: No

3. Has the statistical analysis been performed appropriately and rigorously?

Reviewer #1: Yes

Reviewer #3: (No Response)

Reviewer #4: No

4. Have the authors made all data underlying the findings in their manuscript fully available?

Reviewer #1: No

Reviewer #3: Yes

Reviewer #4: Yes

5. Is the manuscript presented in an intelligible fashion and written in standard English?

Reviewer #1: Yes

Reviewer #3: (No Response)

Reviewer #4: Yes

Reviewer #1: (No Response)

Reviewer #3: Thank you for the opportunity to review your manuscript titled "A Deep Learning AI Model for Determining the Relationship Between X-Ray Detectors and Patient Positioning in Chest Radiography." Your work addresses a highly relevant and timely topic, especially as AI continues to influence the optimisation of imaging protocols and the improvement of diagnostic quality in medical imaging.

The manuscript presents a promising application of deep learning to assess patient positioning in relation to X-ray detector alignment a factor critical for image quality and diagnostic accuracy. The use of AI to provide potential decision support or automated feedback to radiographers has clear clinical implications, particularly in standardising chest radiography practices and minimising repeat exposures.

Strengths of the manuscript include:

-A well-motivated research problem with clear clinical relevance.

-Appropriate use of deep learning methodology.

-Potential for integration into clinical workflows for quality assurance or education.

However, there are a few areas that could benefit from further clarification or development:

1.Model Explainability: Consider expanding on how the model's outputs (e.g., heatmaps or classification scores) are interpreted in the context of positioning accuracy.

2.Data Description: Please provide more detail regarding the dataset used, such as the size, diversity of cases (e.g., supine vs. erect exams, if this variation was used), and how positioning quality was labelled.

3.Clinical Validation: It would strengthen the paper to include or suggest a pathway for clinical validation, especially if the model will be used in actual clinical radiography settings.

4.Discussion of Limitations: While the model's performance is promising, a more detailed discussion of its current limitations and generalisability to other institutions or equipment types would enhance transparency.

Overall, the manuscript makes a valuable contribution to the field of AI in radiography and could be further strengthened by addressing the above points. I encourage the authors to refine the manuscript based on the below focused feedback and look forward to seeing this work contribute to improved imaging practices in clinical environments.

Introduction: line 56 -57

Suggestion: In or before this paragraph, include a brief Hx of the ResNet architecture and how it has made a contribution to medical imaging advancements

Study design: line 96 -97

Should this not be in the past tense if it was a completed action?

Materials and methods: line 155

Include what led to the selection of the ResNet18 specifically vs. ResNet34 or other architectures - I am not sure if I have missed in in the manuscript.

In text and Table data

Should this not align with what you have in text because elsewhere there’s spacing or formatting variation (e.g., “Model A”, “ModelA”). In some places you have model and in others modle

Discussion: line 251

Include a limitations subsection

Under future works and outlooks- line 338 Was this not a limitation?

Conclusion:

Reword to include broader implications or future integration with the hospital workflows.

Reviewer #4: The objective of this manuscript is to develop an AI system to precisely determine the spatial relationship between the X-ray detector and the patient during chest radiography. While this is a relevant and clinically significant goal, I have several concerns and suggestions regarding the proposed methodology and the presentation of the work.

1. The manuscript claims to develop an AI system, but it primarily employs relatively simple or already established models. The methodological novelty or technical innovation needs to be clearly demonstrated. A detailed explanation or justification for the choice of models, including an analysis of the model architecture, theoretical basis, and rationale for selecting specific AI algorithms, is essential. In high-stakes medical applications, models require extensive validation, tuning, and interpretability to be considered reliable.

2. The Methods, Results, and Discussion sections are described in broad terms and lack the scientific precision expected in a research manuscript. It would be helpful if the methods were detailed more thoroughly to improve the reproducibility and credibility of the work. Similarly, the Results and Discussion should be supported by clear descriptions, thorough analysis, and well-structured interpretation.

3. The data source section lacks clarity. For example, in line 113, the term “images deemed irrelevant” is vague—please specify the exclusion criteria. The manuscript should also clarify the standards for data inclusion, exclusion, and cleaning. Additionally, the five classification categories are not clearly defined. Similar ambiguities appear throughout the manuscript and should be addressed.

4. In line 146, the manuscript mentions reviewing numerous journals but cites only a few references from the same source. I recommend expanding the literature review to include a broader and more diverse range of high-impact references. Moreover, the novelty of the proposed model remains unclear—it would be helpful if the manuscript clarified how it differs from existing approaches, including reference [17].

5. Please include a more thorough explanation of how the dataset was divided into training, validation, and test sets. If cross-validation or folds were used, a clear description of the process would improve understanding.

6. If there are existing published methods that address similar problems, it would be helpful to include comparisons.

7. Given the clinical implications, I suggest a more comprehensive evaluation strategy. Referring to earlier peer-reviewed studies that have applied validated techniques (before 2022) could strengthen the validation.

8. Is there any limitations of the proposed approach?

9. The manuscript would benefit from careful language editing to enhance clarity and professionalism. For instance, the term “take” in the short title is unclear. Additionally, the phrase “daily chest X-ray” in the abstract, does it refer to routine imaging? Clarification would be helpful.

10. Figures and tables should be clearly labeled and thoroughly explained.

**Do you want your identity to be public for this peer review?** For information about this choice, including consent withdrawal, please see our Privacy Policy

Reviewer #1: No

Reviewer #3: No

Reviewer #4: No

---

## [Author Response · Author response to Decision Letter 2]

12 Aug 2025

We have thoroughly revised the manuscript according to the editors' and reviewers' feedback. The complete manuscript has been uploaded, along with the Response to Reviewers and the Revised Manuscript with Track Changes.We have uploaded a new cover letter and named the file "cover letter3."

---

## [Decision Letter · Decision Letter 2]

10 Sep 2025

A Deep Learning AI Model for Determining the Relationship Between X-Ray Detectors and Patient Positioning in Chest Radiography

PONE-D-24-42489R2

Dear Dr. 司徒,

We’re pleased to inform you that your manuscript has been judged scientifically suitable for publication and will be formally accepted for publication once it meets all outstanding technical requirements.

Kind regards,

Ihssan S. Masad, PhD

Academic Editor

PLOS ONE

Additional Editor Comments (optional):

Reviewer #3:

Reviewers' comments:

Reviewer's Responses to Questions

**Comments to the Author**

Reviewer #3: All comments have been addressed

2. Is the manuscript technically sound, and do the data support the conclusions?

Reviewer #3: Yes

3. Has the statistical analysis been performed appropriately and rigorously?

Reviewer #3: Yes

4. Have the authors made all data underlying the findings in their manuscript fully available?

Reviewer #3: Yes

5. Is the manuscript presented in an intelligible fashion and written in standard English?

Reviewer #3: Yes

Reviewer #3: I would like to thank the authors for carefully considering and addressing the feedback provided on the previous version of the manuscript. The revisions have substantially improved the clarity, methodological rigor, and overall quality of the paper. I appreciate the effort made to incorporate the suggested changes and provide detailed responses to the comments.

**Do you want your identity to be public for this peer review?** For information about this choice, including consent withdrawal, please see our Privacy Policy

Reviewer #3: No

---

## [Editor Report · Acceptance letter]

PONE-D-24-42489R2

PLOS ONE

Dear Dr. Situ,

I'm pleased to inform you that your manuscript has been deemed suitable for publication in PLOS ONE. Congratulations! Your manuscript is now being handed over to our production team.

Kind regards,

on behalf of

Dr. Ihssan S. Masad

Academic Editor

PLOS ONE